# Deep Learning of Determinantal Point Processes via Proper Spectral Sub-gradient

**Tianshu Yu, Yikang Li, Baoxin Li**
Arizona State University
{tianshuy,yikang.li,baoxin.li}@asu.edu

## Abstract

Determinantal point processes (DPPs) is an effective tool to deliver diversity in multiple machine learning and computer vision tasks. Under the deep learning framework, DPP is typically optimized via approximation, which is not straightforward and has some conflicts with the diversity requirement. We note, however, there have been no deep learning paradigms to optimize DPP *directly* since it involves matrix inversion that may result in computational instability. This fact greatly hinders the use of DPP on some specific objective functions where DPP would otherwise serve as a term to measure the feature diversity. In this paper, we devise a simple but effective algorithm to optimize the DPP term directly through expressing with L-ensemble in the spectral domain over the gram matrix, which is more flexible than learning on parametric kernels. By further taking into account additional geometric constraints, our algorithm seeks to generate valid sub-gradients of the DPP term in cases where the DPP gram matrix is not invertible (no gradients exist in this case). In this sense, our algorithm can be easily incorporated with multiple deep learning tasks. Experiments show the effectiveness of our algorithm, indicating promising performance for practical learning problems.

## 1 Introduction

Diversity is desired in multiple machine learning and computer vision tasks (e.g., image hashing (Chen et al., 2017; Carreira-Perpinán & Raziperchikolaei, 2016), descriptor learning (Zhang et al., 2017), metric learning (Mishchuk et al., 2017) and video summarization (Sharghi et al., 2018; Liu et al., 2017)), in which sub-sampled points or learned features need to spread out through a specific bounded space. Originated from quantum physics, determinantal point processes (DPP) have shown its power in delivering such properties (Kulesza et al., 2012; Kulesza & Taskar, 2011b). Compared with other diversity-oriented techniques (e.g., entropy (Zadeh et al., 2017) and orthogonality (Zhang et al., 2017)), DPP shows its superiority as it incorporates only one single metric and delivers genuine diversity on *any bounded space* (Kulesza et al., 2012; Affandi et al., 2013; Gillenwater et al., 2012). Therefore, DPP has been utilized in a large body of diversity-oriented tasks.

In general, sample points from a DPP tend to distribute diversely within a bounded space $\mathcal{A}$ (Kulesza et al., 2012). Given a positive semi-definite kernel function $\kappa : \mathcal{A} \times \mathcal{A} \to \mathbb{R}$, the probability of a discrete point set $\mathcal{X} \subset \mathcal{A}$ under a DPP with kernel function $\kappa$ can be characterized as:

$$\mathcal{P}_\kappa(\mathcal{X}) \propto \det(\mathbf{L}_\mathcal{X}) \tag{1}$$

where $\mathbf{L}$ is a $|\mathcal{X}| \times |\mathcal{X}|$ matrix with entry $\mathbf{L}_{ij} = \kappa(\mathbf{x}_i, \mathbf{x}_j)$ and $\mathbf{x}_i, \mathbf{x}_j \in \mathcal{X}$. $\mathbf{L}$ is called L-ensemble. Note that $\mathcal{A}$ is a continuous space, whereas $\mathcal{X}$ is finite. In the Hilbert space associated with $\kappa$, larger determinant implies larger spanned volume, thus the mapped points tend not to be similar or linearly dependent.

DPP can be viewed from two perspectives: sampling and learning. A comprehensive introduction to mathematical fundamentals of DPP for sampling from a discrete space can be found in Kulesza et al. (2012). Based on this, a line of works has been proposed (Kulesza & Taskar, 2011a; Kang, 2013; Hennig & Garnett, 2016). In this paper, we concentrate on learning DPPs. In learning of DPP, the term $\det(\mathbf{L})$ is typically treated as a singleton diversity measurement and is extended to learning paradigms on continuous space (Chao et al., 2015; Kulesza & Taskar, 2010; Affandi et al., 2014). There are generally two lines of strategies to learn DPPs:

**Approximation.** This type of methods is to convert DPP into a simpler format which can ease and stabilize the computation. *low-rank approximation* proves powerful in easing the computational burden (Gartrell et al., 2017), in which the gram matrix is factorized as $\mathbf{L} = \mathbf{B}\mathbf{B}^\top$ where $\mathbf{B} \in \Re^{n \times m}$ with $m \ll n$. This decomposition can also reduce the complexity which is originally a cubic time of $|\mathbf{L}|$. Kulesza & Taskar (2011b) explicitly expressed the kernel with $\kappa(\mathbf{x}, \mathbf{y}) = \sigma_1 \sigma_2 \delta(\mathbf{x})^\top \delta(\mathbf{y})$, where $\sigma$ measures the intrinsic quality of the feature and $\delta(\cdot)$ is function mapping input $\mathbf{x}$ to a feature space. In this sense, the pairwise similarity is calculated in Euclidean feature space with cosine distance. Elfeki et al. (2019) suggest approximating a given distribution by approximating the eigenvalues of the corresponding DPP. As such, the computation can be eased and become stable. Following this, DPP is also applied on some visual tasks, such as video summarization (Sharghi et al., 2018), ranking (Liu et al., 2017) and image classification (Xie et al., 2017). It can be noted that the approximation is not straightforward for DPP, thus cannot fully deliver the diversity property (e.g. resulting in rank-deficiency).

**Direct optimization.** While the aforementioned methods optimize DPP with specific approximation, a series of efforts also seek to optimize the DPP term *directly* (Gillenwater et al., 2014; Mariet & Sra, 2015; Bardenet & Titsias, 2015). In this setting, the whole gram matrix $\mathbf{L}$ corresponding to the pairwise similarity among features is updated directly, which allows accommodating more flexible feature mapping functions rather than an approximation. Gillenwater et al. (2014) proposed an Expectation-Maximization algorithm to update marginal kernel DPP $\mathbf{K} = \mathbf{L}(\mathbf{L} + \mathbf{I})^{-1}$, together with a baseline K-Ascent derived from projected gradient ascent (Levitin & Polyak, 1966). Mariet & Sra (2015) extended DPP from a fixed-point perspective and Bardenet & Titsias (2015) proposed to optimize DPP upon a lower bound in variational inference fashion. A key problem of such line of works is that the computation is not differentiable, making it difficult to be used in deep learning frameworks.

To the best of our knowledge, there is no previous method incorporating DPP as a feature-level diversity metric in *deep learning*. A key difficulty in doing so is that the calculation of the gradient of $\det(\mathbf{L})$ involves matrix inversion, which can be unstable and inaccurate in GPUs. Though K-Ascent seems to be a naive rule, it still needs explicit matrix inversion in the first step before the projection procedure. This fact greatly hinders the tight integration of DPP with deep networks. Some alternative methods seek to reach diversity under more constrained settings. For example, Zhang et al. (2017) resorted to a global pairwise orthogonality constraint in hyper-sphere and Zadeh et al. (2017) employed statistical moments to measure the diversity. However, compared with DPP, such measurements are unable to fully characterize diversity in an arbitrary bounded space.

In this paper, rather than providing more efficient DPP solvers, we concentrate on delivering a feasible feature-level DPP integration under the deep learning framework. To this end, we revisit the spectral decomposition of DPP and propose a sub-gradient generation method which can be tightly integrated with deep learning. Our method differs from either approximation or direct optimization by introducing a "differentiable direct optimization" procedure, thus can produce genuinely diverse features in continuous bounded space. Our method is stable and scalable to the relatively large dataset with a specific mini-batch sampling strategy, which is verified by several experiments on various tasks.

**Notations:** Bold lower case $\mathbf{x}$ and bold upper case $\mathbf{K}$ represent vector and matrix, respectively. $\det(\cdot)$ and $\mathrm{Tr}(\cdot)$ calculate the determinant and trace of a matrix, respectively. $\mathbf{A} \otimes \mathbf{B}$ is the element-wise product of matrices $\mathbf{A}$ and $\mathbf{B}$. $|\mathcal{X}|$ and $|\mathbf{x}|$ measure the cardinality of a finite set $\mathcal{X}$ and the $\mathrm{L}^2$ length of a vector $\mathbf{x}$, respectively. $\langle \mathbf{x}, \mathbf{y} \rangle$ calculates the inner product of the two vectors. $\mathbf{x} = \mathrm{diag}(\mathbf{X})$ transforms a diagonal matrix $\mathbf{X}$ into its vector form $\mathbf{x}$, and vice versa. We refer "positive semi-definite" and "positive definite" to PSD and PD, respectively. Denote $\Re$ the real numbers.

## 2 BACKGROUND

### 2.1 DETERMINANTAL POINT PROCESS

L-ensemble expression of DPP requires $\mathbf{L}$ to be PSD, whereas kernel expression further constrains $\mathbf{K} \prec \mathbf{I}$ (each eigenvalue of $\mathbf{K}$ is less than 1). A conversion from $\mathbf{L}$ to $\mathbf{K}$ can thus be written as $\mathbf{K} = \mathbf{L}(\mathbf{L} + \mathbf{I})^{-1}$ following the truth $\sum_{\mathcal{X}} \det(\mathbf{L}_{\mathcal{X}}) = \det(\mathbf{L} + \mathbf{I})$, which is the marginal normalization constant given a specific $\mathbf{L}$. While there is always conversion from $\mathbf{L}$ to $\mathbf{K}$, the inverse may not

exist (Kulesza et al., 2012). In practice, one may construct L-ensemble first, then normalize it into a marginal kernel. This fact may give rise to the difficulty of deep networks. Since a conversion from $\mathbf{K}$ to $\mathbf{L}$ might not exist, the network needs carefully adjusting the gradients under specific constraints to ensure the updated $\mathbf{L}$ to be valid. As $\mathbf{L}$ and $\mathbf{K}$ share the same eigenvectors $\mathbf{v}_i$, a pair of $\mathbf{L}$ and $\mathbf{K}$ holds the relation:

$$\mathbf{K} = \sum_i \lambda_i \mathbf{v}_i \mathbf{v}_i^\top \quad \Longleftrightarrow \quad \mathbf{L} = \sum_i \frac{\lambda_i}{1 - \lambda_i} \mathbf{v}_i \mathbf{v}_i^\top \tag{2}$$

where $\lambda_i$ is the $i$th eigenvalue. It is seen that such conversion is not straightforward to be directly integrated with deep learning framework. Therefore, we optimize ensemble $\mathbf{L}$ directly in this paper.

## 2.2 GAUSSIAN KERNEL

We briefly introduce Gaussian kernel in this section, which works on Hilbert space with infinite dimension. Mercer's theorem Friedman et al. (2001) ensures the PSD properties when constructing new kernels with existing ones under a specific procedure. Such procedure is also employed in multiple kernel learning paradigms (Affandi et al., 2014; Kulesza & Taskar, 2011b; Chao et al., 2015), which is out of the scope of this paper. A Gaussian kernel is defined as $\kappa(\mathbf{x}_i, \mathbf{x}_j) = \exp\left(-|\mathbf{x}_i - \mathbf{x}_j|^2/\sigma^2\right)$, where $\sigma$ is a controlling parameter. Thus an L-ensemble matrix becomes $\mathbf{L}_{ij} = \kappa(\mathbf{x}_i, \mathbf{x}_j)$. According to the definition, $\mathbf{L}_{ii} = 1$ and for any element in the matrix we have $\mathbf{L}_{ij} \in (0, 1]$. With Gaussian kernel, we have a nice property $0 \leq \det(\mathbf{L}) \leq 1$. This can be easily verified by applying geometric inequality to the eigenvalues of $\mathbf{L}$. Although not tight, this property shows that the determinant value with Gaussian kernel is bounded. This fact inspires one version of our algorithm detailed in the next section. Throughout this paper, our discussion is based on the Gaussian kernel unless specified.

## 3 METHOD

Given vectorized inputs $\mathbf{I}_i \in \mathbb{R}^h$ where $i = 1, ..., n$, our goal is to learn a map $f$ such that the features $\mathbf{x}_i = f(\mathbf{I}_i)$ can spread out within a bounded feature space $\mathbf{x}_i \in \mathcal{S}$. Hereafter we refer *space* to an Euclidean bounded space (e.g., $[-1, 1]^d$) without loss of generality. Given any loss function $J$, the chain rule of gradient involving DPP is written as:

$$\Delta J = \frac{\partial J}{\partial \det(\mathbf{L})} \frac{\partial \det(\mathbf{L})}{\partial \mathbf{L}} \frac{\partial \mathbf{L}}{\partial \mathbf{X}} \tag{3}$$

where $\mathbf{X}$ refer to the features before DPP layer. While calculating $\partial J/\partial \det(\mathbf{L})$ and $\partial \mathbf{L}/\partial \mathbf{X}$ is straightforward, the main difficulty lies on the calculation of $\partial \det(\mathbf{L})/\partial \mathbf{L}$. We will discuss the calculation of this term under two case: 1) When the inversion $\mathbf{L}^{-1}$ can be stably obained, we will derive the gradient of DPP $\det(\mathbf{L})$ on Sec 3.1; When $\mathbf{L}$ is not invertible or $\mathbf{L}^{-1}$ is difficult to calculate, we give the procedure to handle the case by generating valid sub-gradient in Sec 3.2. Since our objective is to diverse features, $\det(\mathbf{L})$ will serve as a (partial) objective term to be directly maximized.

---

**Algorithm 1** DPPSG

**Input: K**, tolerance $\Delta$; **Output:** $\bar{\mathbf{K}}$
$\mathbf{U}\boldsymbol{\Lambda}\mathbf{U}^\top \leftarrow \mathbf{K}$
$(\sigma_1, ..., \sigma_n) \leftarrow \mathrm{diag}(\mathbf{K})$
**for** $i$ in $\{1, ..., n\}$ **do**
    **if** $\sigma_i < \Delta$ **then**
        $\sigma_i \leftarrow \Delta$
    **end if**
**end for**
$\hat{\boldsymbol{\Lambda}} \leftarrow \mathrm{diag}(\sigma_1, ..., \sigma_n)$
$\hat{\mathbf{K}} \leftarrow \mathbf{U}\hat{\boldsymbol{\Lambda}}\mathbf{U}^\top$
$\bar{\mathbf{K}} \leftarrow \hat{\mathbf{K}} - \mathbf{K}$

---

**Algorithm 2** DPPSG*

**Input: K**, tolerance $\theta$; **Output:** $\bar{\mathbf{K}}$
$\mathbf{U}\boldsymbol{\Lambda}\mathbf{U}^\top \leftarrow \mathbf{K}$
$(\sigma_1, ..., \sigma_n) \leftarrow \mathrm{diag}(\mathbf{K})$
$\mathbf{b} = (1 - \theta)(\sigma_1, ..., \sigma_n) - \theta(1, ..., 1)$
$\hat{\boldsymbol{\Lambda}} \leftarrow \mathrm{diag}(\mathbf{b})$
$\hat{\mathbf{K}} \leftarrow \mathbf{U}\hat{\boldsymbol{\Lambda}}\mathbf{U}^\top$
$\bar{\mathbf{K}} \leftarrow \hat{\mathbf{K}} - \mathbf{K}$

---

### 3.1 Derivation of gradient

With kernel $\kappa$, a DPP regularization term seeks to maximize the possibility of a feature configuration $\mathbf{x}_i$, $i = 1, ..., n$. As this possibility is proportional to $\det(\mathbf{L})$, the objective is $\max \det(\mathbf{L})$. This can become a regularization term where diversity is required. Thus with a general loss function $L_G$, our aim is to solve $\min L_G - \lambda_1 \det(\mathbf{L})$, with the controlling parameter $\lambda_1 \geq 0$. For the time being, we assume that kernel matrix $\mathbf{L}$ is invertible (we will discuss the case when $\mathbf{L}$ is not invertible in the next section), hence $\mathbf{L}^{-1}$ exists. Without loss of generality, we discuss the gradient of the determinant equipped with Gaussian kernel. For other kernels the derivation is analogous. According to Eq (**??**), $\mathbf{L}_{ij}$ can be further factorized as:

$$\mathbf{L}_{ij} = \exp\left(-\frac{\sum_l (\mathbf{x}_{il} - \mathbf{x}_{jl})^2}{\sigma^2}\right) \tag{4}$$

where $\mathbf{x}_{ij}$ is the $j$th dimension of feature $\mathbf{x}_i$. Using chain rule, the derivative of $\det(\mathbf{L})$ w.r.t. $\mathbf{x}_{il}$ can be written as:

$$\frac{\partial \det(\mathbf{L})}{\partial \mathbf{x}_{il}} = \det(\mathbf{L}) \operatorname{Tr}\left(\mathbf{L}^{-1} \frac{\partial \mathbf{L}}{\partial \mathbf{x}_{il}}\right) \tag{5}$$

where on the $ij$th position of $\frac{\partial \mathbf{L}}{\partial \mathbf{x}_{il}}$ the corresponding element is:

$$\left(\frac{\partial \mathbf{L}}{\partial \mathbf{x}_{il}}\right)_{ij} = \exp\left(-\frac{|\mathbf{x}_i - \mathbf{x}_j|^2}{\sigma^2}\right)\left(-\frac{2(\mathbf{x}_{il} - \mathbf{x}_{jl})}{\sigma^2}\right) \tag{6}$$

Eq (6) can be more compactly expressed as:

$$\frac{\partial \mathbf{L}}{\partial \mathbf{x}_{il}} = \mathbf{L} \otimes \mathbf{M}^{(il)} \tag{7}$$

where $\mathbf{M}^{(il)}$ is such a matrix that, except for the $i$th column and row, all resting elements are $0$s. Besides, the $ij$th and $ji$th elements of $\mathbf{M}^{(il)}$ are both $-\frac{2(\mathbf{x}_{il} - \mathbf{x}_{jl})}{\sigma^2}$. In summary, Eq (5) can be simplified as:

$$\frac{\partial \det(\mathbf{L})}{\partial \mathbf{x}_{il}} = \det(\mathbf{L}) \operatorname{Tr}\left(\mathbf{L}^{-1}\left(\mathbf{L} \otimes \mathbf{M}^{(il)}\right)\right) \tag{8}$$

To ease the computation and fully utilize the chain rule in deep learning architecture, we peel the DPP loss into two layers, and the corresponding gradient product can be expressed as:

$$\left(\frac{\partial \det(\mathbf{L})}{\partial \mathbf{L}}\right) \cdot \left(\frac{\partial \mathbf{L}}{\partial \mathbf{x}}\right) \tag{9}$$

While we can use existing package to obtain $\partial \mathbf{L}/\partial \mathbf{x}$ reliably, the way to stably calculate $\partial \det(\mathbf{L})/\partial \mathbf{L}$ becomes essential. We will detail in the next section once the term is hard to calculate.

### 3.2 Proper Spectral sub-gradient for back-propagation

The calculation of the gradient $\partial \det \mathbf{L}/\partial \mathbf{L}$ involves computing the inverse matrix $\mathbf{L}^{-1}$. However, the kernel matrix $\mathbf{L}$ is not always invertible. This situation happens iff there exists at least a pair of features $\mathbf{x}_i$ and $\mathbf{x}_j$ such that $\mathbf{x}_i = \mathbf{x}_j$. In this case, there exist two identical columns/rows of $\mathbf{L}$ and the $0$ eigenvalue results in the non-invertibility. This phenomenon is sometimes caused by Relu function, which can map different input values onto an identical one. Even when all features are distinct, the numerical precision (typically on float number in GPU) may also lead to failure. We occasionally observed that GPU calculation of $\mathbf{L}^{-1}$ reports error even no eigenvalue is $0$. One may imagine a naive replacement of matrix inverse with the pseudo-inverse, which can be applied on singular matrices. However, pseudo-inverse will keep the zero eigenvalues intact (still rank-deficiency), and the back-propagated gradient will play no part to increase the determinant value (both $0$ before and after updates).

To address this, we first diverge to consider the objective of DPP $\max \det(\mathbf{L})$. Since DPP term seeks to maximize the determinant, for a configuration $\mathbf{L}^{(t)}$ at iteration $t$ with $\det(\mathbf{L}^{(t)}) = 0$, any sufficiently small $\eta$ sufficing $\det(\mathbf{L}^{(t+1)}) > 0$ with $\eta = \mathbf{L}^{(t+1)} - \mathbf{L}^{(t)}$ can be a valid ascending direction. Thus we give the following definition:

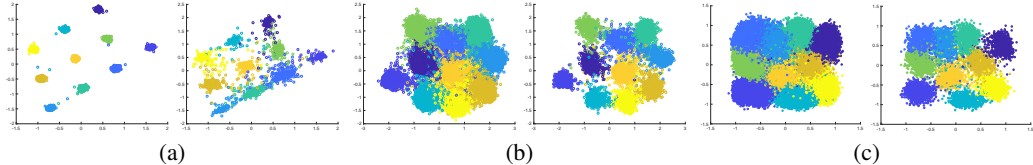

$$\text{(a)} \qquad\qquad\qquad\qquad \text{(b)} \qquad\qquad\qquad\qquad \text{(c)}$$

Figure 1: (a)-(c): feature distribution of MNIST dataset under different settings. Left and right for training and testing samples, respectively. (a) Contrastive loss for metric learning; (b) Contrastive loss + DPP regularization; (c) Contrastive loss + DPP regularization + WGAN regularization. For (c), the features are generally lying in the space $[-1, 1]^2$. Zoom in for better view.

**Definition 3.1.** *Proper Sub-gradient: For a PSD matrix $\mathbf{L}$ such that $\det(\mathbf{L}) = 0$, $\hat{\mathbf{L}}$ is called its proper sub-gradient if $\hat{\mathbf{L}}$ is a sub-gradient and $\det(\mathbf{L} + \alpha\hat{\mathbf{L}}) > 0$ for sufficiently small $\alpha > 0$.*

We see if a proper sub-gradient $\hat{\mathbf{L}}$ can be found at $\det(\mathbf{L}) = 0$, back-propagation procedure in deep learning can consequently perform calculation using $\hat{\mathbf{L}}$. To obtain such $\hat{\mathbf{L}}$, we first note that $\mathbf{L}$ can be eigen-decomposed as following since it is symmetric and PSD:

$$\mathbf{L} = \mathbf{U}\mathbf{\Lambda}\mathbf{U}^\top \tag{10}$$

where $\mathbf{U}$ is the orthogonal eigenvector matrix and $\mathbf{\Lambda}$'s diagonal elements are the corresponding eigenvalues. As $\mathbf{L}$ has zero eigenvalues, the rank of $\mathbf{\Lambda}$ is lower than the dimension of $\mathbf{L}$. We sort all eigenvalue into descending order to $\mathbf{k} = (\sigma_1, ..., \sigma_q, 0, ..., 0)$, where $q < n$. We then employ a simple yet effective amplification procedure by amplifying any eigenvalue smaller than $\Delta$ to $\Delta$. The amplified eigenvalues are now $\bar{\mathbf{k}} = (\sigma_1, ..., \sigma_s, \Delta, ..., \Delta)$, where $s \leq q$. Let the diagonalized amplified eigenvalue matrix be $\bar{\mathbf{\Lambda}}$ (w.r.t. $\mathbf{k}$), then the modified matrix with small positive determinant can be written as:

$$\bar{\mathbf{L}} = \mathbf{U}\bar{\mathbf{\Lambda}}\mathbf{U}^\top \tag{11}$$

Now that $\det(\bar{\mathbf{L}}) = \prod_{i=1}^{q}\sigma_i \prod_{j=q+1}^{n}\Delta > 0$. For any $\epsilon > 0$, we can choose a sufficiently small $\Delta$ such that $\det(\bar{\mathbf{L}}) < \epsilon$. Thus the continuity of this procedure is guaranteed. The difference $\hat{\mathbf{L}} = \bar{\mathbf{L}} - \mathbf{L}$ can be viewed as a proper ascending direction w.r.t. $\mathbf{L}$, as by adding $\hat{\mathbf{L}}$, $\det(\mathbf{L} + \hat{\mathbf{L}})$ becomes above $0$ as well as arbitrarily small. It is trivial to prove that $\hat{\mathbf{L}}$ is a sub-gradient on a neighbor of $\mathbf{L}$, thus $\hat{\mathbf{L}}$ is also a proper sub-gradient sufficing Definition 3.1. This procedure is summarized in Algorithm 1 and is termed as **DPPSG**. Intuitively, once encountering an identical or too close feature pair $\mathbf{x}_i$ and $\mathbf{x}_j$, this procedure tries to enhance the diversity by separating them apart from each other.

Inspired by geometric inequality, we provide an improved version of the algorithm taking into account the property of Gaussian kernel. First it easy to show that the function $\prod_i \sigma_i$ is concave in the feasible set $\sum_i \sigma_i = n$ (diagonal of Gaussian gram matrices are 1s, thus trace is $n$) and the maximal objective is reached out iff $\sigma_i = 1$. Therefore, any point $\mathbf{b} = (1 - \theta)(\sigma_1, ..., \sigma_n) + \theta(1, ..., 1)$ will increase the objective $\prod_i \sigma_i$. By letting $\theta$ being a small value, the proper sub-gradient becomes $\mathbf{U}\text{diag}(\mathbf{b} - \sigma)\mathbf{U}^\top$, where $\sigma = (\sigma_1, ..., \sigma_n)$. This version of update differs from **DPPSG** as it generates sub-gradients under geometric constraints. The method is summarized in Algorithm 2 and is termed as **DPPSG\***.

During implementation, the irregularity of $\mathbf{L}$ is examined to determine whether to adopt a normal back-propagation (in Sec 3.1) or sub-gradient (in Sec 3.2). This can be done by verifying if the determinant value in the forward pass is less than a pre-defined small value $\beta$. This proper sub-gradient based back-propagation method can be used to integrate to deep learning framework with other objectives involving matrix determinant. We emphasize that our method is different from the line of gradient-projection based methods, such as K-Ascent. While projection-based methods calculate the true gradient then project it back to a feasible set, our methods *generate proper sub-gradient directly*. Without explicitly computing matrix inversion, sub-gradients, in this case, is more feasible for deep learning framework.

**Mini-batch sampling** We employed a balanced sampling strategy for each mini-batch. Assuming the batch size is $n$ and there are $c$ classes in total, in each mini-batch the distribution of samples generally follows the whole training sample distribution on $c$ classes. This strategy is considered to utilize the intrinsic diversity of the original data. Besides, mini-batch sampling can constrain the overhead of DPP computation depending only on the batch size, which can be viewed as a constant in practice.

### 3.3 BOUNDING THE FEATURES WITH WASSERSTEIN GAN

Practically, the features are always required to lie in a bounded space. This is essential in some applications as a bounded space is more controllable. Especially, sometimes one may demand that the features should suffice to a pre-defined distribution $\mathcal{P}$. This bounding requirement is crucial to the objective of DPP since maximizing determinant tends to draw feature points infinitely apart from each other. A naive method to achieve this is to truncate the features or using barrier functions. However, these methods will result in irregularly dense distribution on the learned feature space boundary. To overcome this issue, we employ Wasserstein GAN (WGAN) Arjovsky et al. (2017) to enforce the features mapped to a specific distribution $\mathcal{P}$. As we do not focus on WGAN in this paper, readers are referred to Arjovsky et al. (2017) for more details.

To this end, we randomly sample $n^1$ points $\bar{\mathbf{x}}_i$ from the distribution $\mathcal{P}$ under balanced sampling, which are treated as positive samples. The generator $f(\cdot)$ takes a feature as input and outputs the corresponding embedding. Denote the discriminator $h(\cdot)$ (which is also the mapping from input to feature). Then the WGAN loss for discriminator is:

$$L_W = \mathbb{E}_{\bar{\mathbf{x}} \sim \mathcal{P}}\left[h(\bar{\mathbf{x}})\right] - \mathbb{E}_{\mathbf{I} \sim p(\mathbf{I})}\left[h(f(\mathbf{I}))\right] \tag{12}$$

According to the Arjovsky et al. (2017), we incorporate the generator loss $L_C = -\mathbb{E}[h(f(\mathbf{I}))]$ into general loss $L_G$ and obtain $L = L_G - \lambda_1 L_D + \lambda_2 L_C$, where $\lambda_1 > 0$ and $\lambda_2 > 0$ are controlling parameters and $L_D$ is the DPP term. In general, the second and third losses serve as regularization. While the DPP term $L_D$ makes the points spread out over the whole space, the WGAN term $L_W$ enforce the points to be under a distribution $\mathcal{P}$. These two terms are set to negative as we seek to maximize them. In the implementation, $L_W$ and $L$ are trained alternatively.

| | mAP-$k$(%) | | | |
|---|---|---|---|---|
| $k$ | 10 | 20 | 50 | 100 |
| baseline | 63.90 | 62.87 | 61.43 | 58.65 |
| DPPSG | 67.22 | 67.45 | 65.82 | 62.78 |
| DPPSG* | 67.94 | 68.73 | 66.32 | 62.75 |
| DPPSG+WGAN | 68.07 | 69.34 | 66.19 | 63.40 |
| DPPSG*+WGAN | 69.14 | 70.32 | 68.04 | 64.58 |

Table 1: Retrieval performance on MNIST.

| | mAP-$k$(%) | | |
|---|---|---|---|
| b-size | 10 | 20 | 50 |
| 200 | 30.50 | 31.28 | 32.49 |
| 300 | 30.78 | 32.27 | 32.29 |
| 400 | 31.44 | 33.46 | 33.51 |
| 500 | 33.97 | 34.49 | 35.38 |

Table 2: Impact of batch size (b-size) in CIFAR-100 100 classes.

## 4 EXPERIMENTS

In this section, we conduct two experiments. One is about metric learning and image hashing on MNIST and CIFAR to verify the effectiveness of our method, while another is for local descriptor retrieval task based on HardNet (Mishchuk et al., 2017). For the first test.

### 4.1 VERIFICATION TEST

**MNIST** This simple dataset is suitable to reveal the geometric properties of the features on various tasks. We test the image retrieval task equipped with contrastive loss $L_C = \sum_{\mathcal{L}(i)=\mathcal{L}(j)} (\mathbf{x}_i - \mathbf{x}_j)^2 + \alpha \sum_{\mathcal{L}(i)\neq\mathcal{L}(j)} \max\left(\mu - (\mathbf{x}_i - \mathbf{x}_j)^2, 0\right)$ under Gaussian DPP regularization, where $\mathcal{L}(i)$ indicates the label of the $i$th feature and $\mathbf{x}_i$ is the learnt feature. We employ a simple network structure for

---

[1]We choose $n$ as the batch size for simplicity. One can change this value.

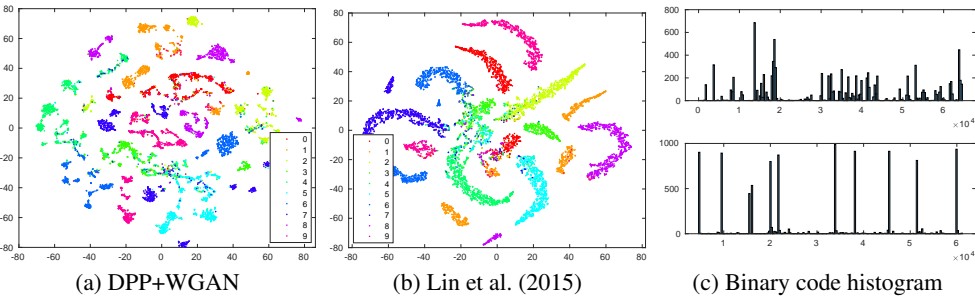

|                |                |                        |
| :------------: | :------------: | :--------------------: |
| (a) DPP+WGAN   | (b) Lin et al. (2015) | (c) Binary code histogram |

Figure 2: Visualization of 16–bit hashing code results on CIFAR–10. For (c) the histograms up and down correspond to DPP+WGAN and Lin et al. (2015), respectively. Zoom in for better view.

MNIST. This network consists of 3 convolutional layers (Conv) followed by 2 fully connected layers (FC). Batch normalization (Ioffe & Szegedy, 2015) is applied on each layer. The filter number of each Conv are $32$, $32$ and $64$, respectively. The sizes of the filter are identically $5 \times 5$. For the first Conv, we employ maxpooling. For the other 2 Convs, average pooling is adopted. The dimensions of the last FCs are 200 and 2 (for 2D visualization).

The performance can be found in Table 1 and the feature distribution is visualized in Figure 1. From Table 1, it is observed that the performance on retrieval task can be enhanced by adding the DPP and WGAN regularization terms. We see that DPP term can enhance the retrieval performance by avoiding feature points from concentrating too much. In this sense, the learned map around the separating boundary can be much smoother. As retrieval task typically requires the existence of top-$k$ inter-class samples rather than concentrating property, the DPP term is more preferable. In Figure 1(c), we see that the feature points generally fall into the pre-defined space $[-1, 1]^2$. The utility of such space is high without sacrificing the retrieval performance. Typically, DPPSG* is slightly superior to DPPSG. Thus in the following test, we only report the performance under DPPSG* setting (termed as DPP* for short).

**CIFAR–10 image hashing** We conduct image hashing on CIFAR–10 which seeks to produce binary code for images. To this end, we follow the binary hashing code generation procedure in Lin et al. (2015) which is activated by a Sigmoid function. The number of neurons in the second last layer equals to the number of bits of the hashing codes. It is anticipated that DPP regularization can enhance the utility in binary code space since the code can spread out[2]. We test two lengths of binary code (12 and 16). We visualize the 16–bit feature distribution using TSNE (Maaten & Hinton, 2008) in Fig. 2 (a) and (b), and the binary code histogram comparison in (c). The quantitative results are summarized in Table 2. As Lin et al. (2015) jointly solve binary code generation and classification, we report both retrieval performance (mAP) and classification performance (Acc). We see our method can significantly enhance the binary space utility while keeping the performance almost intact.

**CIFAR–100 metric learning** We employ all the convolutional layers in VGG-19 (Simonyan & Zisserman, 2014) as the base and discard its final fully connected layers. Thus the output size of this base VGG-19 network is $1 \times 1 \times 512$. We concatenate 3 fully connected layers with ReLU activation on each after that with dimensions 512, 100 and 20, respectively. Contrastive loss is applied on the 20-dimensional space. We train the whole network from scratch. Aside from mAP, we also report top-$k$ average precision (Precision-$k$) and the Wasserstein distance to the pre-defined distribution (Gap to $\mathcal{P}$). The performance on coarse (20 classes) and fine (100 classes) levels can be found in Table 3. In either setting, we see that DPP+WGAN significantly outperform the baseline. Thus we infer that the DPP term can serve as a regularization not only for the feature itself but also for the smoothness of the mapping. Since the DPP term avoids the features from concentrating too much, the learned mapping should also be from a smoother function family.

**Batch size VS. performance** We study how batch size influences the performance with DPP regularization. To this end, we report the performance on CIFAR-100 100-class retrieval with different batch sizes. The results are shown in Table 2. Generally, with larger batch size, the algorithm can reach out better mAP. We note the computational efficiency of DPP sub-gradients is high, which adds

---

[2]Higher binary code space utility can enhance the hashing speed and save the storage.

very slight overhead (even with 500 batch size) to each iteration of common back-propagation under contrastive loss, which can be neglected.

| | mAP-$k$ (%) | | | Precision-$k$ (%) | | | Gap to $\mathcal{P}$ |
|---|---|---|---|---|---|---|---|
| $k$ | 10 | 20 | 50 | 10 | 20 | 50 | |
| On coarse (20) classes | | | | | | | |
| Baseline | 6.98 | 6.74 | 6.82 | 9.44 | 9.36 | 9.35 | – |
| DPP* | 45.35 | 48.09 | 48.74 | 55.62 | 53.04 | 51.08 | – |
| DPPW* | 47.30 | 52.18 | 54.37 | 60.60 | 58.44 | 57.39 | 0.046 |
| On fine (100) classes | | | | | | | |
| Baseline | 17.98 | 18.24 | 18.06 | 23.21 | 23.47 | 22.76 | – |
| DPP* | 28.43 | 28.49 | 28.37 | 35.18 | 34.79 | 33.26 | – |
| DPPW* | 30.50 | 31.28 | 32.49 | 40.15 | 40.76 | 38.36 | 0.032 |

Table 3: Metric learning performance on CIFAR–100 dataset with course (20) and fine (100) classes.

| | mAP-$k$ (%) | | | Acc |
|---|---|---|---|---|
| k | 50 | 100 | all | |
| 12–bit | | | | |
| DCH | 82.9 | 83.9 | 85.9 | 83.5 |
| DPP | 81.7 | 81.9 | 81.7 | 89.9 |
| 16–bit | | | | |
| DCH | 84.9 | 85.4 | 86.7 | 92.0 |
| DPP | 83.9 | 83.7 | 82.9 | 91.5 |

Table 4: Image hashing on CIFAR–10. "Acc" is the classification accuracy.

## 4.2 LOCAL DESCRIPTOR RETRIEVAL

This test utilizes the UBC Phototour dataset (Brown & Lowe, 2007), which consists of three subsets (Liberty, Notre Dame, and Yosemite) with around 400k $64 \times 64$ local patches for each. We follow the protocol in Mishchuk et al. (2017) to treat two subsets as the training set and the third one as the testing set. As each pair of matched image patches includes only two patches, there is no need to apply balanced sampling in this test. We simply add DPP regularization term to the objective of state-of-the-art algorithm HardNet (Mishchuk et al., 2017). The batch size is 512. We report FPR (false positive rate) and FDR (false discovery rate) following Mishchuk et al. (2017); Han et al. (2015). Results are summarized in Table 5. Several baselines are selected for comparison (i.e. SIFT (Lowe, 1999), MatchNet (Han et al., 2015), TFeat-M (Balntas et al., 2016), L2Net (Tian et al., 2017) and HardNet (Mishchuk et al., 2017)). As the authors improved HardNet after the NeurIPS submission, we also compare with the latest version (termed as HardNet+). We only conduct our method under DPPSG* setting and name our method HardDPP. We see that with DPP regularization, the performance of HardNet can be further enhanced. Note that in HardNet there is no WGAN integrated as the mapped features lie in the surface of a hyper unit sphere. While the sampling strategy of HardNet emphasizes the embedding behavior near the margin, DPP regularization can further focus on global feature distribution.

| Training Testing | Notre + Yose Lib | | Lib + Yose Notre | | Lib + Notre Yose | | Mean FDR | FPR |
|---|---|---|---|---|---|---|---|---|
| SIFT | | 29.84 | | 22.53 | | 27.29 | | 26.55 |
| MatchNet | 7.04 | 11.47 | 3.82 | 5.65 | 11.6 | 8.7 | 7.74 | 8.05 |
| TFeat-M | 7.39 | 10.31 | 3.06 | 3.8 | 8.06 | 7.24 | 6.47 | 6.64 |
| PCW | 7.44 | 9.84 | 3.48 | 3.54 | 6.56 | 5.02 | | 5.98 |
| L2Net | 3.64 | 5.29 | 1.15 | 1.62 | 4.43 | 3.3 | | 3.24 |
| HardNet | 3.06 | 4.27 | 0.96 | 1.4 | 3.04 | 2.53 | 3.0 | 2.54 |
| HardNet+ | 1.47 | 2.67 | 0.62 | 0.88 | 2.14 | 1.65 | | 1.57 |
| HardDPP | **1.21** | **2.17** | **0.58** | **0.70** | **1.79** | **1.32** | **1.31** | **1.17** |

Table 5: Performance of UBC Phototour comparison. Notre, Yose and Lib are short for "Notre Dame", "Yosemite" and "Liberty", respectively. Following HardNet Mishchuk et al. (2017), we report FPR at true positive rate at 95%. The best results are in **bold**.

## 5 CONCLUSION

In this paper, we investigated the problem of learning diverse features via a determinantal point process under deep learning framework. To overcome the instability in computing the gradient which involves the matrix inverse, we developed an efficient and reliable procedure called proper spectral sub-gradient generation. The generated proper sub-gradient can replace the true gradient and performs well in applications. We also considered how to constrain the features into a bounded space, since in such a way one can ensure the behavior of the network more predictable. To this

end, we further incorporated Wasserstein GAN into our framework. Together, DPP+WGAN showed significant performance on both some common criteria and feature space utility.

ACKNOWLEDGMENTS

The work was supported in part by a grant from ONR. Any opinions expressed in this material are those of the authors and do not necessarily reflect the views of ONR.

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

## A  APPENDIX

### A.1  PARAMETER SET-UP

**MNIST** Some parameters are set as follows: $\alpha = 5$, $\lambda_1 = 10^3$, $\lambda_2 = 10^6$, margin $\mu = 0.8$, variance for Gaussian kernel $\sigma = 0.2$ and $\Delta = 10^{-7}$. During the training, the batch size is set to 200. In each iteration of DPP and WGAN training, we uniformly sample $2,000$ adversarial points from the space $[-1,1]^2$. We adopt RMSprop and the learning rate is $10^{-4}$ for all tests. In the testing stage, we sample $2,000$ points from $[-1,1]^2$ and calculate the Wasserstein distance with all the testing samples. This procedure is conducted 10 times and the mean distance is reported.

**CIFAR–10 image hashing** The parameters in the hashing related experiments are used as following: variance for Gaussian Kernel $\sigma = 2$, the coefficients for the loss term of DPP is $\lambda_1 = 10^2$ and for the loss term of discriminator and generator in WGAN is 10 and 1 respectively. The batch size is set to $500$ and the learning rate is initialized to $0.01$ with a changing rate of $0.1$ at every 150 epoch. The total number of epoch is set to 350 and we adopt the Adam optimizer to update our model.

**CIFAR–100 metric learning** The parameter setting is as follows: $\alpha = 1$, $\lambda_1 = 10^3$, $\lambda_2 = 10^3$, margin $\mu = 0.8$, variance for Gaussian kernel $\sigma = 0.2$ and $\Delta = 10^{-6}$. The rest of the settings are the same as those of MNIST test.

### A.2 CRITERIA PRECISION-$k$ AND MAP-$k$

For image retrieval task, we adopt the top-$k$ mean average precision (abbreviated as mAP-$k$) to evaluate the performance. We also present the top-$k$ average precision (abbreviated as Precision-$k$), which is calculated as:

$$\text{Precision}(b_j)@K = \frac{\sum_{i=1}^{K} \mathbb{I}(b_j)P(b_j)@i}{\sum_{i=1}^{K} \mathbb{I}(b_j)} \tag{13}$$

where $b$ is the corresponding class and $\mathbb{I}$ is the indicator function:

$$\mathbb{I}(b) = \begin{cases} 1 & \text{if b is a true positive} \\ 0 & \text{if b is a false positive} \end{cases} \tag{14}$$

Thus mAP-$k$ is the reweighted version of Precision-$k$:

$$\text{mAP}(b_j) = \frac{\sum_{j=1}^{N} \text{Precision}(b_j)@K}{N} \tag{15}$$

### A.3 OVERHEAD OF DPP

Calculating SVD or matrix inversion on a large number of features can be time consuming. However, in our setting, we employed a common practice in deep learning – mini-batch – to avoid such computation on a whole batch. We can conclude that mini-batch strategy can limit the computational cost such that the extra overhead of DPP is only dependent on the batch size (thus other parts of the networks have no impact on this overhead). Therefore, although the complexity of our method is $\mathcal{O}(n^3)$, $n$ only corresponds to the batch size rather than whole sample number in our setting, which is much more manageable in practice. We report the average overhead comparison on CIFAR-10 hashing task with varying batch sizes (100, 200, 250, 400 and 500) on a GTX 1080 GPU as in Table 6 (time in seconds):

| batch size | 100 | 200 | 250 | 400 | 500 |
|---|---|---|---|---|---|
| overhead-all | 0.175 | 0.263 | 0.308 | 0.411 | 0.493 |
| overhead-DPP | 0.011 | 0.029 | 0.033 | 0.042 | 0.056 |

Table 6: Overhead of a single batch and a DPP calculation on CIFAR-10 hashing task with varying batch size. Time is in seconds.

where "overhead-all" and "overhead-DPP" refer to the average time cost (s) for a single batch on all the computation and only DPP computation (both forward and backward), respectively. We can conclude that, compared to other computation, the extra overhead of DPP is small (even in a simple network as CIFAR-10 hashing). Besides, a batch size up to 500 is considered to be sufficient in most of the applications. In practice, we did not employ any trick to reduce such overhead (since it is out of the papers focus) but simply utilized standard functions provided by Pytorch.

### A.4 SOME BASELINE DETAILS

For **MNIST** verification test, we employed a simple backbone. The structure of the backbone is {conv_1($5 \times 5$)+maxpool+conv_2($5 \times 5$)+avepool+conv_3($5 \times 5$)+avepool+fully_con1(200-d)+fully_con2(2-d)+fully_con3(10-d)+contrastive_loss} . We add DPP and WGAN regularization to the features at "fully_con2" layer, which is 2-dimensional thus better for visualization.

For **CIFAR-10 image hashing** task, we employ the same network structure as a high-cited method DCH (Lin et al., 2015). DPP and WGAN loss is applied on the second last fully connected layer (the dimension of this layer corresponds to the length of digits in the hashing code).

For **CIFAR-100 metric learning** task, we employ all the convolutional layers of VGG19, concatenated with 3 fully connected layers (with 512, 100 and 20 dimension). DPP and WGAN loss, together with contrastive loss, is applied on the final fully connected layer (20-dimension). The network is trained from scratch without any pre-training.

## A.5 DEGRADATION ON CIFAR-10 IMAGE HASHING

For the performance degradation with DPP on hashing task, we can take Figure 2 (c) as an example to explain. We see that original DCH features concentrate on several digits (generally 10 digits corresponding to 10 classes), while DPP features diffuse to almost the whole discrete space. In this sense, if one retrieves the $k$-th closest hashing code, DCH can find the hashing code with a small searching radius. However, one has to greatly enlarge the search radius for $k$-th closest code in DPP feature space since the distribution is much more even. In this sense, DPP will inevitably causes degradation since large searching radius will more likely to reach a code in other class. Therefore, we think "utility vs mAP" is an intrinsic conflict and needs to reach a trade-off.

