# OpenReview forum: "Deep Learning of Determinantal Point Processes via Proper Spectral Sub-gradient"
_ICLR.cc/2020/Conference — Accept (Poster)_

### Official Review · AnonReviewer1 · 2019-10-20
**Official Blind Review #1**

**Rating:** 8

**Review:**

Determinantal Point Processes (DPPs) are statistical models that allow efficient sampling of diverse solutions - a problem that is hard with most machine learning modeling frameworks.  However, learning diverse features via DPPs with deep learning frameworks is challenging due the instability in computing the gradient which involves a matrix inverse operation. Better marriage between deep learning and DPPs is an important problem as diversity is crucial in many machine learning problems (even beyond computer vision - the experimental domain of this paper - e.g. in machine translation and document summarization).

The authors of this paper make several important contribution to this problem. First, they develop an effective sub gradient procedure, proper spectral subgradient generation,  that can replace the true gradient and empirically demonstrate its effectiveness in computer vision applications.  Then, they describe a method for constraining the DPP features into a bounded space to facilitate network predictability. When incorporating Wasserstein GAN into their framework they show performance gains in computer vision tasks: metric learning, image hashing and local descriptor
retrieval task based on HardNet.

The paper is overall clearly written, but in some places it is not well edited (to the level that is should be carefully proofread). For example, "For the first test." just before 4.1, and "showed significance performance" in the last sentence of the paper. I strongly encourage the authors to carefully edit the paper before publication so that their nice work will be properly presented.



**Experience Assessment:**

I have published one or two papers in this area.

**Review Assessment: Checking Correctness Of Derivations And Theory:**

I assessed the sensibility of the derivations and theory.

**Review Assessment: Checking Correctness Of Experiments:**

I assessed the sensibility of the experiments.

**Review Assessment: Thoroughness In Paper Reading:**

I read the paper at least twice and used my best judgement in assessing the paper.

---

> ### Author Response · Authors · 2019-11-07
> **response to review#1**
>
> Thank you for your overall positive comments and for appreciating the contributions of our work. We will revise our paper to improve its readability.

---

### Official Review · AnonReviewer3 · 2019-10-21
**Official Blind Review #3**

**Rating:** 3

**Review:**

Summary: the authors introduce a method to learn a deep-learning model whose loss function is augmented with a DPP-like regularization term to enforce diversity within the feature embeddings.

Decision: I recommend that this paper be rejected. At a high level, this paper is experimentally focused, but I am not convinced that the experiments are sufficient for acceptance.

****************************
My main concerns are as follows:

- Many mathematical claims should be more carefully stated. For example, the authors extend the discrete DPP formulation to continuous space. It is not clear to me, based on the choice of the kernel function embedding, that the resulting P_k(X) is a probability (Eq. 1). If it is not (using a DPP-based formulation as a regularizer does not require a distribution), the authors should clarify that fact; more generally, the authors should be more careful throughout the paper (for example, det=0 if features are proportional, not necessarily equal; the authors inconsistently switch between DPP kernel L and marginal kernel K throughout computations.)

- The authors do not describe their baselines for several experiments. In tables 1, 2, 3, the baseline is never described (I assume it's the same setup without regularization); I did not find a description of DCH (Tab 4) in the paper (Deep Cauchy Hashing?). The mAP-k metric should also be defined. Furthermore, the authors do not report standard deviations for their experiments.

- A key consideration when using DPPs is their compulational cost: most operations involving them require SVD (which seems to be used in this work), matrix inversion, and often both. This, unsurprisingly, limits the applications of DPPs, and has driven a lot of research focused on improving DPP overhead. I would like to see more discussion in this paper focused on to which extent the DPP's computational overhead remains tractable, and which methods were used (if any) to alleviate the computational burden.

- Finally, the paper itself appears to be somewhat incomplete: sentences are missing or incomplete (Section 4), and numbers are missing in some tables (Table 5).


***********************
Questions and comments for the authors:

- When computing the proper sub-gradient, are you computing the subgradient as inv(L + \hat L)?

- You state that by avoiding matrix inversion, your method is more feasible. However, it seems like your method requires SVD, which is also O(N^3); could you please provide more detail for this?

- Could you report number of trials and standard deviations for your experiments?

- Do you have any insight into why DPPs do more poorly than the DCH baseline in Table 4 for mAP-k metrics?

- You might be able to save space by decreasing the space allocated to the exponentiated quadratic kernel.


**Experience Assessment:**

I have published in this field for several years.

**Review Assessment: Checking Correctness Of Derivations And Theory:**

I carefully checked the derivations and theory.

**Review Assessment: Checking Correctness Of Experiments:**

I assessed the sensibility of the experiments.

**Review Assessment: Thoroughness In Paper Reading:**

I read the paper thoroughly.

---

> ### Author Response · Authors · 2019-11-07
> **response to review#3**
>
> Overall we understand the reviewer’s concern about the experiments's convincingness. While we think this is partly due to the fact our framework is very novel and the problem setting has rarely been tested before. We think the results and numbers in the tables and figures themselves are self-evident to show the usefulness of our method. Furthermore, we would like to to follow reviewer's suggestion to add more discussion and explanation to the results, in the final version to improve the readability of the paper. We give more detailed response as follows.
>
> Complexity: Indeed, as the reviewer stated, calculating SVD or matrix inversion on a large number of features can be time consuming. However, in our setting, we employed a common practice in deep learning - mini-batch - to avoid such computation on a whole batch. We think this also exactly shows the usefulness to combine DPP objective directly in deep learning as they can benefit to each other. We can conclude that mini-batch strategy can limit the computational cost such that the extra overhead of DPP is only dependent on the batch size (thus other parts of the networks have no impact on this overhead). Therefore, although the complexity of our method is O(n^3), n only corresponds to the batch size (rather than whole sample number) in our setting, which is tractable in practice. We report the average overhead comparison on CIFAR-10 hashing task with varying batch sizes (100, 200, 250, 400 and 500) on a GTX 1080 GPU as follows (time in seconds):
>
> b_size	                  100	  200	  250	  400	  500
> Overhead all	         0.175	0.263	0.308	0.411	0.493
> Overhead DPP	0.011	0.029	0.033	0.042	0.056
>
> where “overhead all” and “overhead DPP” refer to the average time cost (s) for a single batch on all the computation and only DPP computation (both forward and backward), respectively. We can conclude that, compared to other computation, the extra overhead of DPP is small (even in a simple network as CIFAR-10 hashing). Besides, a batch size up to 500 is considered to be sufficient in most of the applications. In practice, we did not employ any trick to reduce such overhead (since it is out of the paper’s focus) but simply utilized standard functions provided by Pytorch. We will add the overhead discussion in the revised version.
>
> Math: In our method, we only optimize the L-ensemble L rather than kernel K. Thank the reviewer for pointing out our misuse of L and K. To clarify, all Ks since Section 3 (Method) are Ls. We will clarify this point in the final paper..
>
> P_k(X) in Eq (1) is the probability. We think this equation is proper since we used “proportional” sign instead of “equal” sign (=).
>
> As we use Gaussian kernel throughout the paper, proportional features will not lead to det=0 in this case (one can think about an example with only 2 feature points, in which Gaussian kernel will lead to non-0 det value). det=0 under proportional features will occur with linear kernel (but we are not using it in our method).
>
> Baselines: DCH corresponds to the method in [1]. We will detail the baseline description in the appendix. We report the common criteria mAP-k and precision-k as in a large body of hashing and metric learning papers (e.g., [1][2]), under which the deviation is not well defined. We would be happy to report deviation if the reviewer or someone can provide any reference on how to calculate deviation under these criteria.
>
> Misc: When we calculate a proper sub-gradient, we do NOT calculate inv(L + \hat L). Proper sub-gradient is calculated using \bar L – L (2nd line below Eq (12)), where \bar L is obtained by adjusting the eigenvalues of L (Eq (11)). This procedure corresponds to Definition 1, where a proper sub-gradient is just to guarantee a positive increment of determinant value under a small positive \alpha. In general, we can increase the determinant value each time by stepping towards a proper sub-gradient direction. This matches the objective to maximize the determinant of DPP.
> For the performance degradation with DPP on hashing task, we can take Figure 2(c) as an example to explain. We see that original DCH features concentrate on several digits, while DPP features diffuse to almost the whole discrete space. In this sense, if one retrieves the k-th closest hashing code, DCH can find the hashing code with a small searching radius. However, one has to greatly enlarge the search radius for k-th closest code in DPP feature space since the distribution is much more even. In this sense, DPP will inevitably causes degradation since large searching radius will more likely to reach a code in other class.
> We trained the networks for each task 3-5 times and observed almost no obvious variation of the performance. So we did not report the deviation w.r.t. multiple trials.
>
> [1] Lin, Kevin, et al. "Deep learning of binary hash codes for fast image retrieval." CVPRW. 2015.
> [2] Liu, Haomiao, et al. "Deep supervised hashing for fast image retrieval." CVPR. 2016.

---

### Official Review · AnonReviewer2 · 2019-10-24
**Official Blind Review #2**

**Rating:** 6

**Review:**

The authors present an approach to optimize determinantal point processes directly (by gradient descent, instead of . via approximations), so that diversity could be modeled in objective functions for deep learning systems. The approach taken is to express the DPP term as an L-ensemble in the spectral domain over the gram matrix. They also generate sub-gradients for cases where the gradient does not exist (when the gram matrix is not invertible).

The approach is interesting, and the problem (modeling of diversity constraints) seems important. They have experimental results (on metric learning and image learning tasks) to show that optimization with the DPP + wasserstein  gan constraint (to ensure features lie in a bounded space) result in better quality. However, they do not discuss the performance (comptuational) impact, or compare their approach to other approximation based systems (such as the approach of Elfeki et al).

**Experience Assessment:**

I do not know much about this area.

**Review Assessment: Checking Correctness Of Derivations And Theory:**

I assessed the sensibility of the derivations and theory.

**Review Assessment: Checking Correctness Of Experiments:**

I assessed the sensibility of the experiments.

**Review Assessment: Thoroughness In Paper Reading:**

I read the paper at least twice and used my best judgement in assessing the paper.

---

> ### Author Response · Authors · 2019-11-07
> **response to review#2**
>
> Thank you for your comments and especially for your appreciation of the importance of the work. Regarding the impact on computational complexity, we present the details in 'response to review#3'. We did not compare our method to [Elfeki et al] because their method is built upon a specific pre-given distribution. In our case, we do NOT assume any property of the feature distribution except for a ‘diverse’ description. Besides, our method focuses more on the optimization (especially working on a full-rank DPP under deep learning framework) side, which has not been addressed to our best knowledge.

---

### Decision · Program_Chairs · 2019-12-19

**Decision:**

Accept (Poster)

**Comment:**

Most reviewers seems in favour of accepting this paper, with the borderline rejection being satisfied with acceptance if the authors take special heed of their comments to improve the clarity of the paper when preparing the final version. From examination of the reviews, the paper achieves enough to warrant publication. Accept.